# Boosting Photovoltaic Performance in Organic Solar Cells by Manipulating the Size of MoS_2_ Quantum Dots as a Hole-Transport Material

**DOI:** 10.3390/nano11061464

**Published:** 2021-06-01

**Authors:** Kwang Hyun Park, Sunggyeong Jung, Jungmo Kim, Byoung-Min Ko, Wang-Geun Shim, Soon-Jik Hong, Sung Ho Song

**Affiliations:** 1Division of Advanced Materials Engineering, Kongju National University, Cheonan-si 32588, Korea; recite14@gmail.com (K.H.P.); jsk71317@gmail.com (S.J.); qudals3920@gmail.com (B.-M.K.); hongsj@kongju.ac.kr (S.-J.H.); 2Nano Hybrid Technology Research Center, Korea Electrotechnology Research Institute, 12 Jeongiui-gil, Seongsan-gu, Changwon-si 51543, Korea; jungmokim@keri.re.kr; 3Department of Chemical Engineering, Sunchon National University, 255 Jungang-Ro, Suncheon-si 57922, Korea; wgshim@scnu.ac.kr

**Keywords:** quantum dot, transition metal dichalcogenide, hole-transport layer, polymer solar cells, conventional structure

## Abstract

The design of photoactive materials and interface engineering between organic/inorganic layers play a critical role in achieving enhanced performance in energy-harvesting devices. Two-dimensional transitional dichalcogenides (TMDs) with excellent optical and electronic properties are promising candidates in this regard. In this study, we demonstrate the fabrication of size-controlled MoS_2_ quantum dots (QDs) and present fundamental studies of their optical properties and their application as a hole-transport layer (HTL) in organic solar cells (OSCs). Optical and structural analyses reveal that the as-prepared MoS_2_ QDs show a fluorescence mechanism with respect to the quantum confinement effect and intrinsic/extrinsic states. Moreover, when incorporated into a photovoltaic device, the MoS_2_ QDs exhibit a significantly enhanced performance (5/10-nanometer QDs: 8.30%/7.80% for PTB7 and 10.40%/10.17% for PTB7-Th, respectively) compared to those of the reference device (7.24% for PTB7 and 9.49% for PTB7-Th). We confirm that the MoS_2_ QDs clearly offer enhanced transport characteristics ascribed to higher hole-mobility and smoother root mean square (R_q_) as a hole-extraction material. This approach can enable significant advances and facilitate a new avenue for realizing high-performance optoelectronic devices.

## 1. Introduction

There is enormous interest in two-dimensional van der Waals crystals possessing tunable optical bandgaps and unique electrical properties. As a result, transitional metal dichalcogenides (TMDs, MX_2_; M = Mo and W; X = S, Se, and Te) have gained considerable attention regarding electronic and optoelectronic devices [1,2,3,4]. Their utilization as electron/hole-transport materials to replace inorganic oxides (e.g., MoO_3_, WO_3_, NiO_x_, ZnO_x_, and V_2_O_5_) and poly(3,4-ethylenedioxythiophene): poly(styrenesulfonate) (PEDOT:PSS) has led to a significant enhancement of the power conversion efficiency (PCE) in bulk-heterojunction (BHJ) solar cells. This enhancement has been achieved because they have clear advantages, such as suitable work function values and a widely tunable band gap; they are also easy to process [5,6,7]. Gu et al. demonstrated the fabrication of a MoS_2_ film via deposition of evaporated MoO_3_ and assessed its performance as a hole-transport layer (HTL) in inverted organic solar cells (OSCs)—its efficiency reached ~8.11% [8]. Yun et al. and Le et al. reported the work function modulation of MoS_2_ nanosheets through a p-/n- doping range of 3.5–4.8 eV and application as an HTL in OSC [9,10,11]. Liu et al. studied a solution-processing method ascribed to the exfoliation of MoS_2_ via the addition of a hydrophilic surfactant (hexadecyltrimethylammonium chloride), obtaining an efficiency of ~7.26% [12]. Despite this significant achievement, intensive studies are still necessary to attain higher efficiency and operational stability through the manipulation of energy level alignment and wettability between layers in OSCs.

Quantum dots (QDs) in OSCs are promising candidates due to their band gap tunability, high absorption coefficient, and multiple exciton generation [13,14,15]. The bandgap can be tuned easily by adjusting the size and composition of QDs. This makes them suitable as electron/hole materials in the OSCs, because the mismatches between energy levels or interfacial defects in planar structures largely cause an undesirable recombination loss of the photogenerated charge carriers, as well as instability regarding the device performance [16,17,18]. Li et al. reported a significant improvement in PCE through CuInS_2_ QDs with sequential coatings of a CdS layer, obtaining efficiencies of 4.20% and 5.38%, respectively [19]. Recently, Pan et al. reported a photovoltaic efficiency of ~7% by using CuInS_2_/ZnS core–shell QDs [20]. More recently, Lajafi et al. reported the synergistic effect of TMD-based QDs in photovoltaic devices, which evidently suppress interfacial charge recombination between layers, and obtained the enhanced power conversion efficiency (PCE) of ~20.11% through low-cost and low-temperature processing [21]. Xing et al. demonstrated UV-ozone (UVO)-treated MoS_2_ QDs with a tunable work function as the hole-transport layer in P3HT- and PTB7-Th-based organic solar cells and achieved significantly enhanced performance [22]. Inspired by this contribution [23,24], we argue that tailoring the energy level of TMD-based QDs through size controlling and optimizing band alignment still remain a challenge for realizing high-efficiency OSCs. Our approach can pave the way for further improving the performance of energy-harvesting devices, and the current different-sized MoS_2_ QDs can be immediately applicable in energy-harvesting systems suffering from similar problems.

Herein, we demonstrate a simple and effective route for improving PCE by using size-controlled MoS_2_ QDs as the HTL in OSCs. The MoS_2_ QDs were fabricated via the formation of a MoS_2_/potassium-sodium tartrate (KNaC_4_H_4_O_6_·H_2_O) intercalation compound, the exfoliation of MoS_2_ QDs with a random size distribution in the selected solvents, and size sorting into 5- and 10-nanometer MoS_2_ QDs. From photoluminescence (PL), time-resolved PL (TRPL), and transient absorption spectroscopy measurements, the as-prepared MoS_2_ QDs demonstrated clear characteristic peaks for the MoS_2_ QDs (5 and 10 nm), ascribed to the quantum confinement effect and intrinsic/extrinsic states, respectively. Moreover, the MoS_2_ QDs were evaluated as a hole extraction layer in the OSCs, and they played a key role in attaining high performance in the OSCs (5-nanometer QD for PTB7 and PTB7-Th; PCE: 8.32% and 10.57%, *J*_SC_: 19.2 and 20.1 mA cm^−2^, *V*_OC_: 0.73 and 0.80 V, and *FF*: 59% and 65%, respectively). Overall, the results suggest that scientific strategy through TMD-based QDs (5 and 10 nm MoS_2_ QDs) can provide noteworthy progress for high performance in optoelectronics.

## 2. Materials and Methods 

### 2.1. Preparation of MoS_2_ QDs

Transition metal dichalcogenide (MoS_2_, Sigma–Aldrich Chemicals Co, Miamisburg, OH, USA) and potassium sodium tartrate (Sigma–Aldrich Chemicals Co, Miamisburg, OH, USA) were chosen to fabricate MoS_2_-based quantum dots (5 and 10 nm). The MoS_2_ powder (20 mg) and potassium sodium tartrate (200 mg) were mixed by grinding under N_2_ atmosphere. Then, the grinded homogeneous mixtures were moved into the homemade reactor and kept at 250 °C for 24 h. The MoS_2_ interaction compound was deposited into the selected solvents for fabricating the MoS_2_ QDs. After the purification process of the randomly dispersed MoS_2_ QDs by a 10 KD MWCO dialysis tube, the size sorting of the QDs was conducted by the centrifuge filtration method (10,000 NMWL, Amicon Ultra-15). Then, the 5- and 10-nanometer QDs dispersed in the distilled water were obtained, and the QD solutions were freeze-dried. Finally, the 5- and 10-nanometer QD powders were obtained.

### 2.2. Device Fabrication

A conventional device was selected to evaluate the performance of PTB7:PC_71_BM and PTB7-Th:PC_71_BM-based OSCs with 5 and 10 nm MoS_2_ QDs introduced as the hole-transport layer. The device architecture was glass/ITO/PEDOT:PSS/MoS_2_ QDs/photoactive materials/Al. First, ITO was coated on the glass and cleaned using deionized water, acetone, and isopropyl alcohol, then the substrates were dried in an oven. The PEDOT:PSS on the ITO was spin-casted at 4000 rpm for 1 min and baked at 150 °C for 30 min. Then, 5- and 10-nanometer MoS_2_ QDs were spin-coated on the PEDOT:PSS/ITO substrates at 4000 rpm for 1 min, with varying numbers of spin-castings (1, 3, 5, and 10 times), and were baked at 150 °C for 30 min. The substrate was moved into the glovebox under N_2_ environment. The p-type materials (PTB7 and PTB7-Th: 12 mg mL^−1^) and n-type material (PC_71_BM: 40 mg mL^−1^) were dissolved in a mixture solvent of chlorobenzene/1,8-diiodooctane (97/3 vol%). After the mixing of p-type/n-type materials (1/1.7 wt%), the blends were spin-coated at 900 rpm for 2 min onto substrates deposited on different-sized MoS_2_ QDs. Finally, Al (100 nm) counter electrode was thermally evaporated under vacuum (~10^−6^ Torr), which defined the device area of 13 mm^2^. Device measurements were conducted in the glovebox by a Xenon arc lamp solar simulator. *J−V* characteristics were measured under AM 1.5G illumination (100 mW cm^−2^) with a Keithley 2635 A source measurement unit. The vertical carrier mobility was measured by the space-charge-limited current (SCLC) method [25,26], and the device architecture was hole-only (ITO/PEDOT:PSS/hole-transport layer (5 nm/10 nm MoS_2_ QDs)/photoactive materials/Au). Hole mobility was calculated by the Mott–Gurney relation. EQE measurement was performed under ambient conditions using a PV Measurements QEX7 Solar Cell QE Measurement System (PV Measurements, Inc., Washington, DC, USA).

### 2.3. Characterizations

The optical properties of the MoS_2_ QDs were measured by UV–Vis spectrophotometer (Agilent Carry 5000, Santa Clara, CA, USA), and the morphology of the QD on the PEDOT:PSS/ITO substrate was analyzed by an Agilent 5500 scanning probe microscope (SPM) running with a NanoScope V controller. Structural properties of the MoS_2_ QDs were also analyzed by Raman spectroscopy (LabRAM HR UV/Vis/NIR, excitation at 514 nm). High-resolution transmission electron microscopy (HR-TEM, Tecnai G2 F30, Hillsboro, OR, USA) was measured after the droplet of the QDs suspension on TEM grid. The photoluminescence (PL), time-resolved PL, and excitation-wavelength-dependent PL behavior measurements were conducted at room temperature using a 325-nanometer He-Cd continuous-wave (CW) laser, a mode-locked femtosecond-pulsed Ti:sapphire laser (Coherent, Chameleon Ultra II, Santa Clara, CA, USA), and monochromatic light from a 300 W Xenon lamp, respectively. Work function of the MoS_2_ QDs was measured by ultraviolet photoelectron spectroscopy (He I (21.2 eV) discharge lamp, ESCALAB 250Xi, Thermo Fisher Scientific, Kyoto, Japan) under ultrahigh vacuum (<10^−10^ Torr).

## 3. Results and Discussion

Figure 1a shows the overall synthetic procedure for fabricating the size-controlled MoS_2_ QDs. The MoS_2_ powder (20 mg) was mixed with potassium sodium tartrate (200 mg) under N_2_ atmosphere, placed into the reactor, and heated at 250 °C for 24 h. Then, the compound was cleaved into randomly dispersed MoS_2_ QDs in the selected solvent (distilled water, DI, Appendix A) and the aqueous solution was thoroughly dialyzed by a 10 KD MWCO dialysis membrane in DI, until a neutral pH was reached. Finally, the size-sorting of the MoS_2_ QDs was conducted via centrifugation in filters (8000 and 10,000 NMWL, Amicon Ultra-15) to 5 and 10 nm lateral-sized MoS_2_ QDs at 10,000 rpm for 10 min (bottom (5 nm) and top (10 nm)) in the centrifuge tube. The experimental details are further elaborated in the experimental section. Raman spectroscopy was performed for the bulk MoS_2_, 5 and 10 nm MoS_2_ QDs to determine their structural properties (shown in Figure 1b). Two characteristic peaks are observed at ~382 and ~406 cm^−1^, corresponding to E_2g_ and A_1g_. The E_2g_ and A_1g_ peaks of the as-prepared MoS_2_ QDs exhibit a blueshift of ~2 cm^−1^ and a redshift of ~3 cm^−1^, respectively, and their distances between the E_2g_ and A_1g_ peaks are approximately ∆ 20 and ∆ 21 cm^−1^, respectively. These results indicate that the thicknesses of the 5- and 10-nanometer MoS_2_ QDs evidently decreased to a mono/few-layer thickness, which is consistent with the findings of previous studies [27,28]. Figure 1c shows transmission electron microscopy (TEM) images for the overall size distribution of the 5- and 10-nanometer MoS_2_ QDs. From the high-resolution TEM (HR-TEM) images, we can observe that there is a clear feature with ordered lattice fringe spacing (0.27 nm) indexed as (100) facet of the MoS_2_ crystal.

This result is similar to those of previous works [25,29]. Figure 1d exhibits the thickness distributions for the 5- and 10-nanometer MoS_2_ QDs; they exist within a thickness range of 1–3 nm (~65%). Atomic force microscopy (AFM) images and thickness profiles for the MoS_2_ QDs are shown in Appendix A. Size distribution for 5- and 10-nanometer MoS_2_ QDs is presented in Appendix A. The introduction of oxygen or additional elements for MoS_2_ QDs was evaluated by X-ray photoemission spectroscopy (XPS), as shown in Appendix A.

Figure 2a shows digital images of the 5- and 10-nanometer MoS_2_ QDs dispersed in the DI water before (Top) and after (Bottom) excitation under a 365-nanometer ultraviolet (UV) lamp. Depending on the MoS_2_ QD size, the emitting colors of the 5- and 10-nanometer are blue and green, respectively. To further verify optical properties of the different-sized MoS_2_ QDs, we measured ultraviolet–visible (UV–Vis) absorption spectra, PL, and PL excitation (PLE). The bulk MoS_2_ shows two characteristic peaks at ~620 and ~670 nm (Figure 2b) because of A and B excitonic transitions at the k-point of the Brillouin zone, respectively [27]. The calculated optical band gap of the MoS_2_ flakes is ~1.78 eV. In contrast, the as-prepared MoS_2_ QDs have a wide absorption peak ranging from 250–400 nm, indicating the excitonic features of the MoS_2_ QDs.

In addition, the optical absorption peak of the MoS_2_ QDs is clearly blue-shifted, depending on the size decrease and the presence of oxygen functional groups and defects formed on MoS_2_ basal plane and at edges. The PL and excitation-wavelength-dependent PL (PLE) were measured to further clarify the mechanism of luminescence in the 5- and 10-nanometer MoS_2_ QDs (Figure 2c). The PLE spectra of the MoS_2_ QDs exhibit two peaks at 250 nm, originating from the transition between Mo and S (intrinsic states); additionally, a broad shoulder was observed near 300 nm, related to the transition for the oxygen functional groups and defects (extrinsic states; Appendix A).

The PLE peak intensity of 5-nanometer MoS_2_ QDs at 250 nm is higher than that of 10-nanometer MoS_2_ QDs, whereas the PLE peak intensity of 5-nanometer MoS_2_ QDs at 300 nm is lower than that of that of 10-nanometer MoS_2_ QDs. These results are consistent with the UV–Vis results. In the PL under excitation at 310 nm, the 5-nanometer MoS_2_ QDs show a strong peak at ~440 nm (blue emission), whereas the 10-nanometer MoS_2_ QDs have a maximum peak at ~480 nm (green emission); the PL intensity of 5-nanometer MoS_2_ QDs is stronger than that of 10-nanometer MoS_2_ QDs. This result indicates that the as-prepared 5-nanometer MoS_2_ QDs have a high crystal quality within a size of 1~6 nm and that their luminescence mainly originates from the transition of Mo and S atoms, whereas the luminescence of 10-nanometer MoS_2_ QDs show a wide absorption peak ranging from 250~600 nm, depending on the diverse sizes from 9~14 nm and the presence of oxygen functional groups and defects formed on MoS_2_ basal plane as well as at edges.

To elucidate the origin of the PL and the carrier dynamics of the MoS_2_ QDs, measurements of the PL after adjusting the excitation wavelength (λ_ex_) and the TRPL were conducted using a Ti:sapphire laser and a streak camera detector at room temperature, as shown in Figure 3. With increasing the excitation wavelength, the 5-nanometer MoS_2_ QDs demonstrated maximum PL emissions at 250 nm, and the luminescence emission of the 5-nanometer MoS_2_ QDs notably redshifts from 400 to 520 nm with a steep decay of the PL (Figure 3a). However, the 10-nanometer MoS_2_ QDs appear to have a similar PL emission strength in the 250–390 nm excitation range, and their PL redshifts from 460 to 550 nm with decreasing PL intensity (Figure 3b). Depending on the excitation wavelength, our fundamental understanding is not currently sufficient to clearly identify the mechanism for the origin of the PL between the quantum size and surface states. Accordingly, we assessed the TRPL of the different-sized MoS_2_ QDs at a λ_ex_ value of 266 nm (Figure 3c,d). As time passes, the temporal PL spectra of the 5-nanometer MoS_2_ QDs reveal a clear redshift of the PL peak position from 400 to 460 nm (Figure 3c), whereas the 10-nanometer MoS_2_ QDs display a slight shift from 460 to 480 nm (Figure 3d). This indicates that the optically excited carriers in the MoS_2_ QDs at λ_ex_ value of 266 nm transfer from intrinsic to extrinsic states.

Based on the optical properties of the MoS_2_ QDs, we confirm that, as a hole-transport material, the MoS_2_ QDs can provide significant effects in facilitating the transfer and extraction of charge carriers. The conventional device architecture and chemical structure are shown in Figure 4a. Photoactive materials were chosen consisting of poly[(4,8-bis-(2-ethylhexyloxy)-benzo(1,2-b:4,5-*b′*)dithiophene)-2,6-diyl-*alt*-(4-(2-ethylhexyl)-3-fluorothieno[(3,4-*b*]thiophene-)-2-carboxylate-2,6-diyl)] (PTB7) and poly [4,8-bis[(5-2-ethylhexyl)thiophen-2-yl]benzo[1,2-b:4,5-*b′*]dithiophene-*alt*-3-fluorothieno[3,4-*b*]thiophene-2-carboxylate] (PTB7-Th) as donors, and [6,6]-phenyl-C_71_-butyric acid (PC_71_BM) as an acceptor. Performance was then assessed with different-sized MoS_2_ QDs (5 and 10 nm) as an HTL. We measured a control device consisting of a single-junction donor material (PTB7 or PTB7-Th): acceptor (PC_71_BM) (1:1.7 wt%) blend in the conventional OSCs (21.3 mg mL^−1^) in mixed solvent of chlorobenzene (CB):1,8-diiodooctane (DIO) (97:3 vol%) [26,30,31]. Detailed synthesis and device-testing conditions are provided in the methods. Figure 4b shows a band energy diagram of the materials used in this study; the energy level alignment of the MoS_2_ QDs is suitable [21]. UV photoelectron spectroscopy (UPS) measurements were conducted to determine the Fermi level energy (E_F_), work function (W_F_), and valance band (VB). The secondary electron cut-off energies of the He I (21.22 eV) UPS spectra are shown in Figure 4c. The W_F_ of the 5- and 10-nanometer MoS_2_ QDs are observed at energy positions of ~16.6 and ~17.6 eV, corresponding to W_F_ values of ~4.6 and 3.6 eV, respectively. This result indicates n-type doping of intrinsic MoS_2_ QDs ascribed to S-vacancies, impurities, and structural defects. The maximum energies of the VB for the 5- and 10-nanometer QDs were estimated at ~5.25 eV, as shown in the inset of Figure 4c. These results are consistent with previous works [21,32,33]. The MoS_2_ QDs reveal a maximum energy of the VB that is higher than that of the HOMO of the PEDOT:PSS (−5.2 eV). Consequently, MoS_2_ QDs effectively act as an HTL in the PSC structures.

Figure 4d,e exhibits current density–voltage (*J–V*) characteristics of the MoS_2_ QD-based OSC devices, and the photovoltaic parameters are listed in Table 1. The optimized control devices based on PTB7 and PTB7-Th active materials show a PCE of 7.24%, a short-circuit current density (*J*_SC_) of 17.8 ± 0.3 mA cm^−2^, an open-circuit voltage (*V*_OC_) of 0.71 ± 0.01 V, and a fill factor (*FF*) of 56 ± 1% for PTB7, and a PCE of 9.49%, a *J*_SC_ of 18.9 ± 0.2 mA cm^−2^, a *V*_OC_ of 0.80 ± 0.02 V, and an *FF* of 60 ± 2% for PTB7-Th. To assess the synergistic effect of the MoS_2_ QDs as the HTL, the 5- and 10-nanometer-sized MoS_2_ QDs with mono-/few-layer thicknesses were deposited on the PEDOT:PSS/ITO substrate and the device performance was evaluated using the optimized donors (PTB7 or PTB7-Th):acceptor (PC_71_BM) active materials. Introducing the MoS_2_ QDs significantly enhances performance, compared to the control devices. The photovoltaic results of 5-nanometer MoS_2_ QD in PTB7- and PTB7-Th-based OSCs are as follows: PCE of 8.32%, *J*_SC_ = 19.2 ± 0.2 mA cm^−2^, *V*_OC_ = 0.73 ± 0.01 V, *FF* = 59 ± 1% for PTB7 and PCE of 10.57%, *J*_SC_ = 20.1 ± 0.2 mA cm^−2^, *V*_OC_ = 0.80 ± 0.01 V, *FF* = 65 ± 1% for PTB7-Th. Moreover, the hole mobilities for PTB7-based devices fabricated by deposition of 5/10-nanometer QDs were assessed and estimated by the space-charge-limited current (SCLC) method [30,34], as shown in Appendix A and Experimental section. The devices fabricated by the deposition of 5/10-nanometer MoS_2_ QDs show enhanced hole mobilities (*μ*_h_ = 2.65 × 10^−4^ cm^2^ V^−1^ s^−1^ and *μ*_h_ = 1.32 × 10^−4^ cm^2^ V^−1^ s^−1^ for 5 and 10-nanometer MoS_2_ QDs, respectively) compared to that of the control device (*μ*_h_ = 4.2 × 10^−5^ cm^2^ V^−1^ s^−1^ for the control device). Especially, the 5-nanometer MoS_2_ QD leads to the highest level of hole mobility, which is consistent with the enhanced performance ascribed to values of *J*_SC_ and *FF* in the OSCs.

Appendix A exhibits the UV–Vis absorption spectra of the PTB7:PC_71_BM active material/MoS_2_ QDs/PEDOT:PSS/ITO films containing different-sized MoS_2_ QDs (5 and 10 nm) and observing the increase of the overall absorption peaks with addition of MoS_2_ QDs. External quantum efficiency (EQE) characteristics for the PTB7-Th-based devices fabricated after deposition with 5- and 10-nanometer MoS_2_ QDs are provided (Appendix A). The results clearly suggest the dependence of EQE values ascribed to deposition of the MoS_2_ QD and size, which is in good agreement with the data obtained from the *J–V* characteristics.

Figure 5a,b shows the effect of the 5-nanometer MoS_2_ with varying spin-casting/drying cycles on device performance. We further evaluated these photovoltaic devices with different spin-casting/drying cycles (1, 3, 5, and 10 times). Each deposition cycle of the 5-nanometer MoS_2_ QD on the PEDOT:PSS surface investigated in this study shows improvement in device performance in different cycles (Figure 5a,b), and the corresponding photovoltaic parameters are plotted in Appendix A. In particular, the best device performances of PCE (8.32% for PTB7 and 10.57% for PTB7-Th) are obtained after three casting/drying cycles. To further understand enhanced device performance, the surface morphologies of the MoS_2_ QDs deposited on the PEDOT:PSS layer were evaluated by AFM, as shown in Figure 5c. The surface of the PEDOT:PSS/ITO without the MoS_2_ QD has a rough surface with a high root-mean-square (R_q_) value of 2.1 nm (Appendix A). The surface morphologies and R_q_ values exhibit significant differences with increasing spin-casting/drying cycles. With fewer than three spin-casting/drying cycles, the surface morphology exhibits a finer and smoother surface ranging from 0.7–1.4 nm, while after more than five cycles the film morphology becomes rougher, with R_q_ values in the range of 0.9–1.1 nm. Accordingly, MoS_2_ QDs with finer, smoother surfaces are considered to be one of the major factors contributing to the improvement of the corresponding devices.

## 4. Conclusions

We introduce a promising solution-processed method for producing 0D MoS_2_ QDs with different sizes (5 and 10 nm) via an MoS_2_-potassium sodium tartrate compound. We demonstrated these MoS_2_ QDs as a hole-transport layer in the OSCs. The optical results reveal that 5- and 10-nanometer MoS_2_ QDs have clear and differing characteristics regarding absorption and PL; their origins correspond to quantum confinement effect and intrinsic/extrinsic states. Furthermore, MoS_2_ QDs, when used as a hole extraction layer in OSCs, show synergetic effects in achieving high performance, because of their additional photocurrent generation and favorable hole extraction, as well as their modification of the PEDOT:PSS surface. From these results, we confirm that MoS_2_ QDs, when used as an HTL, have great potential for improving the performance of OSCs, and can facilitate new avenues for the fundamental study of organic photovoltaics.

## Figures and Tables

**Figure 1 nanomaterials-11-01464-f001:**
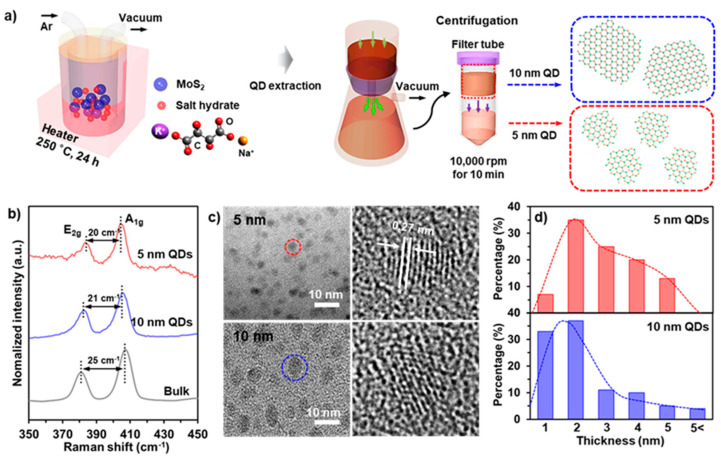
Schematic illustration and characterizations for the MoS_2_ quantum dots. (**a**) fabrication step of the 5- and 10-nanometer MoS_2_ QDs. (**b**) Raman spectra of the bulk MoS_2_ flake, 5- and 10-nanometer MoS_2_ QDs. (**c**) TEM (Left) and HR-TEM (Right) images of the MoS_2_ QDs. (**d**) Thickness distribution of the MoS_2_ QDs.

**Figure 2 nanomaterials-11-01464-f002:**
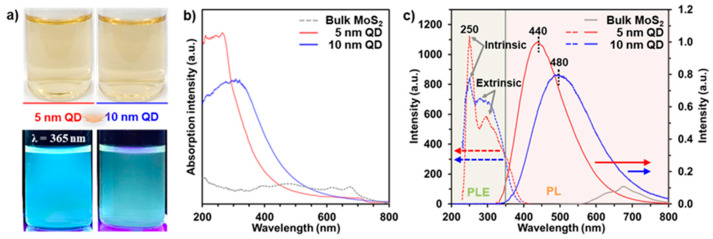
Characterizations for the 5- and 10-nanometer MoS_2_ QDs. (**a**) Digital images of the MoS_2_ QDs before and after emission under excitation of λ = 365 nm. (**b**) UV–Vis absorption spectra of the bulk MoS_2_ flake, 5- and 10-nanometer MoS_2_ QDs. (**c**) PLE and PL spectra of the MoS_2_ QDs under exaction at 440 and 480 nm, respectively.

**Figure 3 nanomaterials-11-01464-f003:**
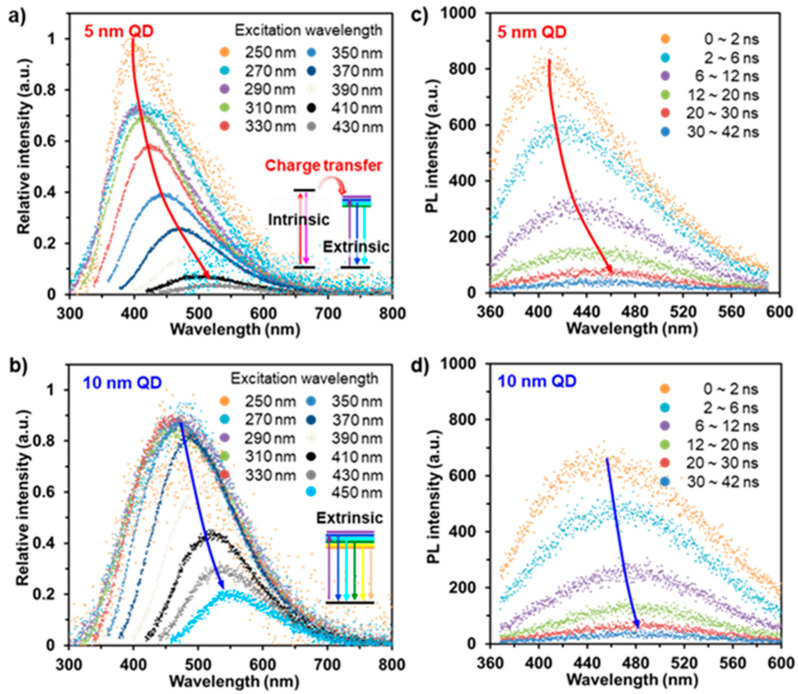
Optical characterizations of carrier dynamics in the 5- and 10-nanometer MoS_2_ QDs. (**a**,**b**) PL spectra of the MoS_2_ QDs with varying excitation wavelength. (**c**,**d**) Time-resolved PL spectra of the MoS_2_ QDs under excitation at 266 nm.

**Figure 4 nanomaterials-11-01464-f004:**
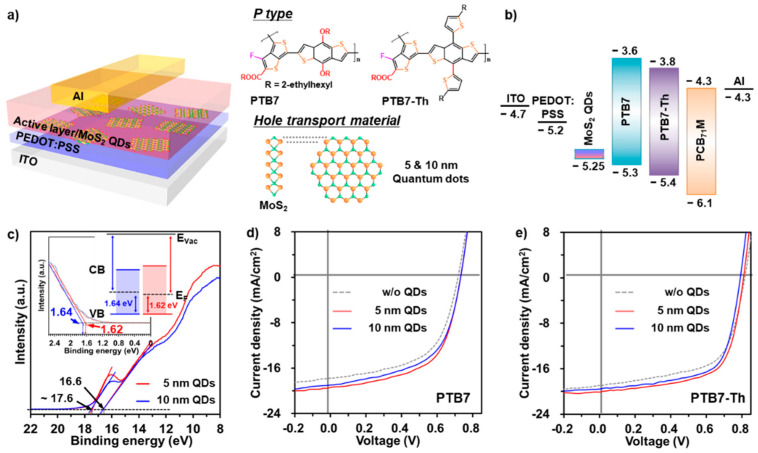
Chemical structures and device characteristics of PTB7 (PTB7-Th):PC_71_BM-based OSCs with/without the 5 -and 10-nanometer MoS_2_ QDs. (**a**) Device structure and molecular structures of donors, acceptor, and MoS_2_ QDs. (**b**) Schematic flat energy band diagram. (**c**) Secondary electron threshold region of He-I UPS spectra of the MoS_2_ QDs. (**d**,**e**) *J–V* characteristics of PTB7-/PTB7-Th-based OSCs.

**Figure 5 nanomaterials-11-01464-f005:**
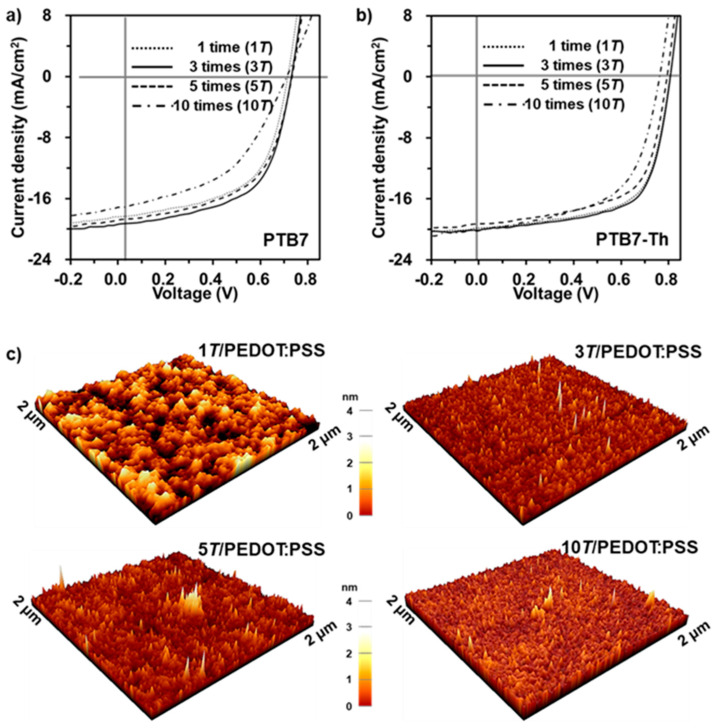
Device characteristics of PTB7 (PTB7-Th):PC_71_BM-based OSCs with varying spin-casting/drying cycle of the 5-nanometer MoS_2_ QDs as hole-transport layer. (**a**,**b**) *J–V* characteristics of PTB7- and PTB7-Th-based OSCs, respectively. (**c**) AFM images of varying spin-casting/drying cycles of the 5-nanometer MoS_2_ QDs.

**Table 1 nanomaterials-11-01464-t001:** Device performance parameters of PTB7 (PTB7-Th):PC_71_BM with varying hole-transport materials based on 5- and 10-nanometer MoS_2_.

Active Materials	Hole-Transport Layer	*J*_SC_ (mA cm^−2^)	*V*_OC_ (V)	*FF* (%)	Avg. PCE (%)
PTB7	w/o MoS_2_ QD	17.8 ± 0.3	0.71 ± 0.01	56 ± 1	7.24
	5 nm MoS_2_ QD	19.2 ± 0.2	0.73 ± 0.01	59 ± 1	8.32
	10 nm MoS_2_ QD	19.0 ± 0.2	0.72 ± 0.01	59 ± 1	7.80
PTB7-Th	w/o MoS_2_ QD	18.9 ± 0.2	0.80 ± 0.02	60 ± 2	9.49
	5 nm MoS_2_ QD	20.1 ± 0.2	0.80 ± 0.01	65 ± 1	10.57
	10 nm MoS_2_ QD	19.6 ± 0.3	0.79 ± 0.01	65 ± 1	10.17

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
