# Peer review of "Boosting Photovoltaic Performance in Organic Solar Cells by Manipulating the Size of MoS2 Quantum Dots as a Hole-Transport Material"

_nanomaterials, 2021, doi:10.3390/nano11061464_

Round 1

Reviewer 1 Report

The authors have presented organic solar cell performance using MoS2 QDs. The organic solar cells have been a great area of research but have inherent issues related to degradation. It is often essential to discuss such issues. The schematic diagram authors presented for MoS2 preparation can be improved and additional information can be included. Excitation dependent/independent emission and associated reasons are greatly described in following paper: 

Light emitting diodes based on carbon dots derived from food, beverage, and combustion wastes 

The authors can describe associated mechanism in their context.

Have authors measured the contact loss?

They should compare some recently published paper with MoS2 and compare the existing efficiency.

Author Response

# Reviewer 1

Comments and Suggestions for Authors

The authors have presented organic solar cell performance using MoS2 QDs. The organic solar cells have been a great area of research but have inherent issues related to degradation. It is often essential to discuss such issues. The schematic diagram authors presented for MoS2 preparation can be improved and additional information can be included. Excitation dependent/independent emission and associated reasons are greatly described in following paper: Light emitting diodes based on carbon dots derived from food, beverage, and combustion wastes

  1. The authors can describe associated mechanism in their context.
  2. Have authors measured the contact loss?
  3. They should compare some recently published paper with MoS2 and compare the existing efficiency.

  1. The authors can describe associated mechanism in their context.

[Response]

We appreciate the reviewer for carefully reviewing our manuscript and providing valuable comments. According to the reviewer’s suggestion, we demonstrated mechanism of transitional metal dichalcogenide (TMDs) and added associated reference in the revised manuscript. Generally, the photoluminescence for carbon dots (CDs) and TMDs QDs is one of the most fascinating features. However, a clear understanding of the PL mechanism still remains unsettled because of effect of multiple excitons ascribed to quantum confinement effect, aromatic structures, emissive traps, oxygen-containing groups, free zigzag sites, and edge defects. In our manuscript, the luminescence of the MoS2 QDs originated from two overlapped spectral bands with intrinsic and extrinsic states and discovered the carrier transfer process from intrinsic to extrinsic states. Also, the size of the MoS2 QDs, crystal defects as well as degree of the oxygen contents evidently induce changes of the PL, PLE and TRPL spectra with different fluorescence colors. In this regard, we confirm that the luminescence of the as-prepared MoS2 QDs consists of two spectrally overlapped bands, corresponding to the transition associated with the intrinsic bands from the crystalline structure of the MoS2 QDs and with extrinsic bands correlated with the oxygen contents and defects formed on the surface or at edges, as shown in in Figure 2 and Figure 3.

Add sentence on the page 5, line 172 of the revised manuscript

In addition, the optical absorption peak of the MoS2 QDs is clearly blue-shifted, depending on the size decrease and the presence of oxygen functional groups and defects formed on MoS2 basal plane and at edges. The PL and excitation wavelength-dependent PL (PLE) were measured to further clarify the mechanism of luminescence in the 5 and 10 nm MoS2 QDs, the (Figure 2c)). The PLE spectra of the MoS2 QDs exhibit two peaks at 250 nm, originating from the transition between Mo and S (intrinsic states); additionally, a broad shoulder was observed near 300 nm, related to the transition for the oxygen functional groups and defects (extrinsic states, Figure S4).

Add sentence on the page 6, line 201 of the revised manuscript

Depending on the excitation wavelength, our fundamental understanding is not currently sufficient to clearly identify the mechanism for the origin of the PL between the quantum size and surface states. Accordingly, we assessed the TRPL of the different-size MoS2 QDs at a λex value of 266 nm (Figure 3c) and d)). As time passes, the temporal PL spectra of the 5 nm MoS2 QDs reveal a clear red-shift of the PL peak position from 400 nm to 460 nm (Figure 3c)), whereas the 10 nm MoS2 QDs display a slight shift from 460 nm to 480 nm (Figure 3d)). This indicates that the optically excited carriers in the MoS2 QDs at λex value of 266 nm transfer from intrinsic to extrinsic states.

  1. Have authors measured the contact loss?

[Response]

We appreciate the insightful comment of the reviewer. Generally, loss of the photogenerated charge carries is influenced by energy barrier and interfacial defect across interfaces. Accordingly, tailoring of energy level optimized at interfaces is one of the most considerable factors to improve device performance in the organic solar cells. In our work, we obtained smoother surface deposited by the 5 nm MoS2 QD, can form better contact with the active layer. Also, the hole mobility of them measured by the space-charge limited current (SCLC) method was evidently enhanced, which is beneficial for charge transport. [Liang et al (1)] These results are quite consistent with them of the photovoltaic performance.

[Reference]

(1) Liang, Z.; Zhang, Q.; Jiang, L.; Cao, G., ZnO cathode buffer layers for inverted polymer solar cells. Energy & Environ. Sci. 2015, 8 (12), 3442-3476.

  1. They should compare some recently published paper with MoS2 and compare the existing efficiency.

[Response]

Thank you for your kind suggestion. According to your comment, we add recently works related with MoS2 and statement in introduction section of the revised manuscript.

Added sentence on the page 2, line 60 of the revised manuscript

Lajafi et al. reported the synergistic effect of TMD-based QDs in photovoltaic devices, which evidently suppress interfacial charge recombination between layers and obtained the enhanced power conversion efficiency (PCE) of ~ 20.11% through low-cost and low-temperature processing.21 Xing et al demonstrated UV−ozone (UVO) treated MoS2 QDs with a tunable work function as hole-transport layer in P3HT and PTB7-Th based organic solar cells and achieved significantly enhanced performance22. Inspired by this contribution23,24, tailoring the energy level of TMD-based QDs through size-controlling and optimizing band alignment still remain a challenge for realizing high-efficiency OSCs. Our approach can pave the way for further improving the performance of energy harvesting devices, and the current different size MoS2 QDs can be immediately applicable in energy harvesting system suffering from similar problems.

[Reference]

  1. Najafi, L.; Taheri, B.; Martín-García, B.; Bellani, S.; Di Girolamo, D.; Agresti, A.; Oropesa-Nunez, R.; Pescetelli, S.; Vesce, L.; Calabro, E., MoS2 quantum dot/graphene hybrids for advanced interface engineering of a CH3NH3PbI3 perovskite solar cell with an efficiency of over 20%. ACS Nano 2018, 12 (11), 10736-10754.
  2. Xing, W.; Chen, Y.; Wang, X.; Lv, L.; Ouyang, X.; Ge, Z.; Huang, H., MoS2 quantum dots with a tunable work function for high-performance organic solar cells. ACS Appl. Mater. Interfaces 2016, 8 (40), 26916-26923.
  3. Sarswat, P. K.; Free, M. L., Light emitting diodes based on carbon dots derived from food, beverage, and combustion wastes. Phys. Chem. Chem. Phys. 2015, 17 (41), 27642-27652.
  4. Kabel, J.; Sharma, S.; Acharya, A.; Zhang, D.; Yap, Y. K., Molybdenum Disulfide Quantum Dots: Properties, Synthesis, and Applications. C 2021, 7 (2), 45.

Reviewer 2 Report

The manuscript by Park et al. reported the fabrication of organic solar cells using MoS2 QDs as hole-transport material. The author reported the enhanced performances as MoS2 QDs offer enhanced transport characteristics. However, the authors should clearly address the following concerns before publication.

  1. Figure 2, PLE stops at ~400 nm for both QDs but absorption still exists, what contributes to the absorption in the range of 400-600 nm?
  2. EQE measurement should be performed and demonstrated in Fig. 4.
  3. The major concern of this paper is that, although the authors didn’t mention, there are appreciable amount of works that use MoS2 QDs to improve the efficiency, including reports with very similar organic solar cell systems (e.g., ACS Appl. Mater. Interfaces 2016, 8, 40, 26916–26923). The authors should properly discuss them in the introduction, and clearly explain the advantages of your approach compared to other reports.

Author Response

# Reviewer 2

Comments and Suggestions for Authors

The manuscript by Park et al. reported the fabrication of organic solar cells using MoS2 QDs as hole-transport material. The author reported the enhanced performances as MoS2 QDs offer enhanced transport characteristics. However, the authors should clearly address the following concerns before publication.

  1. Figure 2, PLE stops at ~400 nm for both QDs but absorption still exists, what contributes to the absorption in the range of 400-600 nm?
  2. EQE measurement should be performed and demonstrated in Fig. 4.
  3. The major concern of this paper is that, although the authors didn’t mention, there are appreciable amount of works that use MoS2 QDs to improve the efficiency, including reports with very similar organic solar cell systems (e.g., ACS Appl. Mater. Interfaces 2016, 8, 40, 26916–26923). The authors should properly discuss them in the introduction, and clearly explain the advantages of your approach compared to other reports.

  1. Figure 2, PLE stops at ~400 nm for both QDs but absorption still exists, what contributes to the absorption in the range of 400-600 nm?

[Response]

We are grateful to the reviewer for evaluating our manuscript. We also doubly thank you for your comments of our manuscript and appreciate your concerns about contribution ascribed to absorption in the range of 400 ~ 600 nm. Generally, the MoS2 QDs fabricated by top-down exfoliation process typically show strong optical absorption in the UV region with a tail extending out into the visible range. The UV-Vis spectra of the MoS2 QDs revealed characteristics of size dependent optical absorption and redshifted with increasing the size due to quantum confinement effect. Also, forming of oxygen related functional groups and crystal defects played important role in determining the absorption peak position of MoS2 QDs.[Xu et al (1) and Song et al (2)] A higher degree of surface oxidation can result in more surface defects, resulting in the red-shifted emission. In our manuscript, the MoS2 QDs are prepared with different sizes of 1 ~ 6nm (5 nm MoS2 QDs) and 9 ~ 14nm (10 nm MoS2 QDs) via filtration method by using filter tube and size distribution for 5 and 10 nm MoS2 QDs is shown in Figure S3 in the revised supplementary material. Also, as the size of the MoS2 QDs increased, the oxygen content and defects increased (Figure S4). Therefore, the as-prepared 10 nm MoS2 QDs have a wide absorption peak ranging from 250 ~ 600 nm, depending on the diverse sizes from 9 nm to 14 nm and the presence of oxygen functional groups and defects formed on MoS2 basal plane and at edges.

Revised sentence on the page 5, line 185 of the revised manuscript

This result indicates that the as-prepared 5 nm MoS2 QDs have a high crystal quality within size of 1 ~ 6 nm, and that their luminescence mainly originates from the transition of Mo and S atoms, whereas the luminescence of 10 nm MoS2 QDs show a wide absorption peak ranging from 250 ~ 600 nm, depending on the diverse sizes from 9 nm ~ 14 nm and presence of oxygen functional groups and defects formed on MoS2 basal plane and at edges.

[Reference]

(1) Xu, Q.; Cai, W.; Li, W.; Sreeprasad, T. S.; He, Z.; Ong, W.-J.; Li, N., Two-dimensional quantum dots: Fundamentals, photoluminescence mechanism and their energy and environmental applications. Mater. Today Energy 2018, 10, 222-240.

(2) Song, S. H.; Kim, B. H.; Choe, D. H.; Kim, J.; Kim, D. C.; Lee, D. J.; Kim, J. M.; Chang, K. J.; Jeon, S., Bandgap widening of phase quilted, 2D MoS2 by oxidative intercalation. Adv. Mater. 2015, 27 (20), 3152-3158.

  1. EQE measurement should be performed and demonstrated in Fig. 4.

[Response]

Thank you for your kind suggestion. According to your comment, the external quantum efficiency (EQE) for PTB7-Th based devices deposited by 5 nm and 10 nm MoS2 QDs was measured and the result is shown in Figure S7. We confirm that the EQE values clearly depended on the MoS2 QD sizes, which is in good agreement with the data obtained from the J-V characteristics.

Revised sentence on the page 8, line 265 of the revised manuscript.

External quantum efficiency (EQE) characteristics for the PTB7-Th based devices fabricated after deposition with 5 nm and 10 nm MoS2 QDs are provided (Figure S7). The result clearly suggests the dependence of EQE values ascribed to deposition of the MoS2 QD and size, which is in good agreement with the data obtained from the J-V characteristics.

  1. The major concern of this paper is that, although the authors didn’t mention, there are appreciable amount of works that use MoS2 QDs to improve the efficiency, including reports with very similar organic solar cell systems (e.g., ACS Appl. Mater. Interfaces 2016, 8, 40, 26916–26923). The authors should properly discuss them in the introduction, and clearly explain the advantages of your approach compared to other reports.

[Response]

Thank you for your kind suggestion. According to your comment, we add the statement for recent MoS2 related studies and advantage of our work in introduction section of the revised manuscript.

Add sentence on the page 2, line 62 and 66 of the revised manuscript

Xing et al demonstrated UV−ozone (UVO) treated MoS2 QDs with a tunable work function as hole-transport layer in P3HT and PTB7-Th based organic solar cells and achieved significantly enhanced performance23. Inspired by this contribution24, tailoring the energy level of TMD-based QDs through size-controlling and optimizing band alignment still remain a challenge for realizing high-efficiency OSCs. Our approach can pave the way for further improving the performance of energy harvesting devices, and the current different size MoS2 QDs can be immediately applicable in energy harvesting system suffering from similar problems.

[Reference]

  1. Xing, W.; Chen, Y.; Wang, X.; Lv, L.; Ouyang, X.; Ge, Z.; Huang, H., MoS2 quantum dots with a tunable work function for high-performance organic solar cells. ACS Appl. Mater. Interfaces 2016, 8 (40), 26916-26923.
  2. Kabel, J.; Sharma, S.; Acharya, A.; Zhang, D.; Yap, Y. K., Molybdenum Disulfide Quantum Dots: Properties, Synthesis, and Applications. C 2021, 7 (2), 45.

Round 2

Reviewer 2 Report

The authors have addressed most of the concerns in the revised manuscript. The paper can be published in the present form.